# An objective criterion to evaluate sequence-similarity networks helps in dividing the protein family sequence space

**Bastian Volker Helmut Hornung** [1,2]*, **Nicolas Terrapon** [1,2]*

**1** Aix Marseille Université, CNRS, UMR 7257 AFMB, Marseille, France, **2** INRAE, USC 1408 AFMB, Marseille, France

* bastian.hornung@gmx.de (BVHH); nicolas.terrapon@univ-amu.fr (NT)

## Abstract

The deluge of genomic data raises various challenges for computational protein annotation. The definition of superfamilies, based on conserved folds, or of families, showing more recent homology signatures, allow a first categorization of the sequence space. However, for precise functional annotation or the identification of the unexplored parts within a family, a division into subfamilies is essential. As curators of an expert database, the Carbohydrate Active Enzymes database (CAZy), we began, more than 15 years ago, to manually define subfamilies based on phylogeny reconstruction. However, facing the increasing amount of sequence and functional data, we required more scalable and reproducible methods. The recently popularized sequence similarity networks (SSNs), allows to cope with very large families and computation of many subfamily schemes. Still, the choice of the optimal SSN subfamily scheme only relies on expert knowledge so far, without any data-driven guidance from within the network. In this study, we therefore decided to investigate several network properties to determine a criterion which can be used by curators to evaluate the quality of subfamily assignments. The performance of the closeness centrality criterion, a network property to indicate the connectedness within the network, shows high similarity to the decisions of expert curators from eight distinct protein families. Closeness centrality also suggests that in some cases multiple levels of subfamilies could be possible, depending on the granularity of the research question, while it indicates when no subfamily emerged in some family evolution. We finally used closeness centrality to create subfamilies in four families of the CAZy database, providing a finer functional annotation and highlighting subfamilies without biochemically characterized members for potential future discoveries.

## Author summary

Proteins perform a lot of functions within living cells. To determine their broad function, we group similar amino-acid sequences into families as their shared ancestry argue for shared functionality. That's what we do in the CAZy database, which covers >300 Carbohydrate-Active enZyme families nowadays. However, we need to divide families into

**Data Availability Statement:** New subfamilies are published on the CAZy website, www.cazy.org. High-resolution files and Cytoscape network files have been uploaded to https://doi.org/10.6084/m9.figshare.21671717. All other relevant data are

within the manuscript and its Supporting Information files.

**Funding:** BVHH is supported by the ERA CoBioTech project SYNBIOGAS, which is funded by BBSRC, grant number BB/T011076/1. The funders had no role in study design, data collection and analysis, decision to publish, or preparation of the manuscript.

**Competing interests:** The authors declare that there are no competing interests associated with the manuscript.

subfamilies to provide finer readability into (meta)genomes and to guide biochemists towards unexplored regions of the sequence space. We recently used Sequence Similarity Networks (SSN) to delineate subfamilies in the large GH16 family, but had to entirely rely on expert knowledge to evaluate and take the final decision until now, which is not scalable, not enough automated and less reproducible. To accelerate the construction of protein subfamilies from sequence similarity networks, we present here an investigation of different network properties, to use as indicators for optimal subfamily divisions. The closeness centrality criterion performed best on artificial data, and recapitulates the decisions of expert curators. We used this criterion to divide four more CAZy families into subfamilies and showed that for others no subfamilies exist. We are therefore able to create new protein subfamilies faster and with more reliability.

## Introduction

The amount of newly produced genomic data increases exponentially. This leads to various challenges in how to treat this data, including how to store and analyze it. Another of these challenges is the functional annotation of such data. Ideally, functional annotation is obtained after the experimental demonstration of protein activity, but due to the deluge of new data this is impossible [1,2]. Computational annotation is performed instead in the vast majority of cases. Many methods are available to annotate genomes [2], and most rely on the assignment of sequences to groups of homologs. This is followed by the transfer of the annotation of experimentally characterized proteins to non-characterized group members. In this context, several levels of granularity can be considered from superfamilies to subfamilies. Superfamilies group many remote homologs that display a similar fold but can largely vary in their molecular functions/specificities, only allowing to have a very general annotation for many proteins [3]. In contrast, subfamilies tend to group very few but closely-related sequences, offering a more reliable/precise annotation but only when at least one member has been biochemically characterized. For the delineation of subfamilies, sequence similarity networks (SSNs) have become increasingly popular [4–6]. SSNs consist in running pairwise comparisons (typically BLASTp) between all family members, and select all matching pairs satisfying a given criterion (typically an e-value cutoff, sometimes with coverage requirements) to draw a network in which the connected components would be subfamilies. The main advantage of SSNs compared to phylogeny is their ability to cope with large families for which the alignment computation and results are problematic. Their main limitation is that (i) they require the exploration of the many possible cutoffs and (ii) the critical step for the determination of the optimal solution fully relies on human expertise. While expert knowledge is arguably one of the very principles of science, it lacks the scalability required to deal with nowadays amount of data and the reproducibility for sharing scientific advances. For experts, SSN tools do not necessarily produce human-friendly visualization of the results, making the selection of a suitable network, out of all possible SSNs, a real challenge. As curators of a specialist resource, the Carbohydrate-Active enZyme (CAZy) database, one of our objectives is to enhance the predictive power of our protein annotation, since our community relies on our annotation and expert knowledge to build hypotheses leading to future discoveries [7]. Due to the large amount of CAZy families (currently >300) and the existence of very large and multifunctional families, there is an urgent need to make the subfamily investigations more performant, as well as reproducible. We therefore decided to investigate currently available methods and indicators from which the optimal SSN can easily and reproducibly be derived by expert curators. We investigated various

clustering methods, dimensionality reduction techniques and various network properties, which indicate grouping and spread of nodes within the network. We applied these methods to artificial datasets, as well as to the recently published SSN of glycoside hydrolase family GH16 [8] and to various families from the structure function linkage database [9]. We show that the *closeness centrality* criterion is a fast and relevant indicator for optimal subfamily delineations. It allows the curator to identify a few possibly optimal networks out of often more than a hundred produced in SSN computations. We furthermore show that in some cases multiple levels of subfamilies could be possible, whereas, in others, no subfamilies exist. After determining this as the appropriate method, we generated and analyzed the networks of nine CAZy families, accelerating our efforts in subfamily creation.

## Results

Our goal was to identify an objective criterion to determine optimal SSN e-value cutoffs, given that our assumption is that an optimal cutoff should maximize the number of connections within the subnetworks/subfamilies, without having any connections towards the other sub-networks/subfamilies. We initially investigated various network properties on artificial data and identified *closeness centrality* as the best suited. This criterion was used to reanalyze already published data based on real families of protein sequences, showing overall good agreement. We further compared SSN results to clustering and dimensionality reduction (see S1 Text, supplementary results), and confirmed the better suitability of the SSNs. We therefore applied this criterion to guide the analysis of subfamilies for several CAZy families, glycoside hydrolase family GH16, GH19, GH51, GH45, GH54, GH55, GH68 and GH130 and the poly-saccharide lyase family PL26.

### Closeness centrality is the best indicator on artificial data

Artificial data was generated (see Methods) to simulate a protein sequence family composed of at least one, or two, subfamilies. In real data, such family would appear as a large network at high e-value thresholds, as all members display a detectable similarity to all others, either directly or by transitivity. With the decrease of the e-value cutoff, edges successively disappear leading to the disconnection of single-nodes or subnetworks which should correspond to subfamilies (Fig 1).

Our artificial data thus consisted of *network collections*. Each *collection* is designed to simulate the SSNs of a family for increasing e-value thresholds, one *network* corresponding to one threshold and containing more edges than the previous and less than the next. Hence all networks in the *collection* are constructed upon two identical subnetworks but could be distinguished by the number of edges between the subnetworks (interconnections but no intraconnections), from one to all possible connections. The different *collections* should simulate families with distinct evolutionary properties, from one (the main network without any subfamilies) to many subfamilies, and correspond to the combination of different types of subnetworks (from fully-connected to single-node subnetworks).

We investigated the behavior of several classical network properties like degree centrality, betweenness centrality and closeness centrality on the *artificial* data. The ideal property should reach its global optimum for a network, which maximizes the connections within its subnetworks, and shows no connections to other subnetworks. In our artificial setting, it should indicate as local optima both the networks consisting of (i) all connections between the two subnetworks and (ii) the two disconnected subnetworks. The desired behavior was obtained from *closeness centrality* (with *wf_improved* parameter set to False) (Fig 1). Especially interesting to note is the partially logarithmic behavior of closeness centrality for the lowest number of connections, which thus scores extremely low. *Betweenness centrality* (and its variations, listed

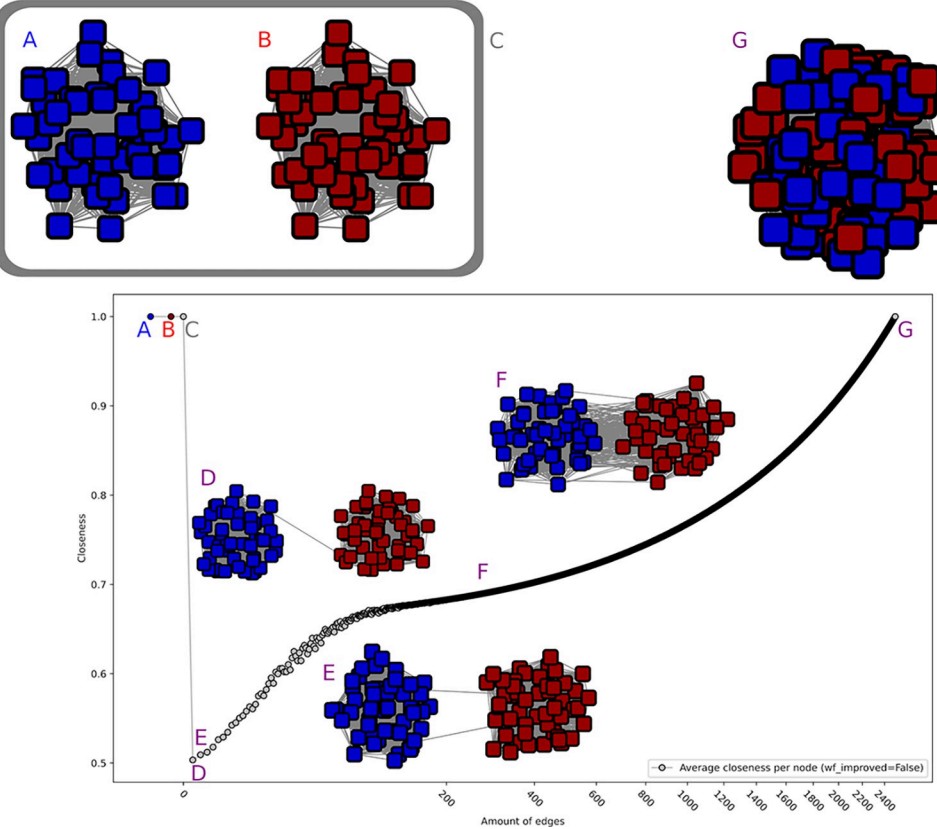

**Fig 1. Illustration of closeness centrality behaviour on artificial data.** Closeness values were averaged over all nodes using two perfect subnetworks of size 50, which sequentially get connected by edges. As it can be seen in A and B, each perfect subnetwork separately reaches a closeness centrality of 1, as well as the combined network C. In all three cases no new edges have been added yet. The value drastically drops when the two networks get connected by a single edge (D), given the average of shortest-path became high (see Closeness Centrality details in Material and Methods). The value then increases with each added edge (E), to intermediate values (e.g. F, 200 edges), and reaches again a maximum value when all edges get connected (G).

in the methods) showed a similar behavior in some network combinations, but was discarded after investigating its behavior in combination with single-node networks. Indeed, *betweeness centrality* weights singles as perfect networks, which is not a desired behavior, since this would favor strongly disconnected networks. A network consisting out of only singles would be evaluated to be as good as a perfect network, which is not what we were looking for. All other criteria behaved either similarly or identical to one of these (one similar to closeness centrality, four similar to betweenness, with betweenness centrality source, load centrality and betweenness behaving identical), were monotonous in their behavior (e.g. two perfect networks with a single connection was evaluated better than a single perfect network; five in total), or not quantitative enough (e.g. asynchronous label propagation indicating one or two subnetworks; three criteria). The remaining four other criteria were (i) the eigenvector centrality and the clustering coefficient, where the former showed parabola-like behavior and the latter an inverted parabola behavior, therefore scoring better with less and more edges than with a medium number of edges, which is the opposite of the desired behavior; (ii) edge load centrality and global reaching centrality, which both partially showed a bimodal distribution. This means that with these algorithms both less and more edges would be favored or disfavored. An overview over all behaviors on two networks with a size of 50 can be seen in S1 Fig.

Based on artificial data results, we therefore opted to use *closeness centrality* (with wf_improved set to False), applied to each subnetwork, as criterion for further evaluation. More details on closeness centrality principle and formula can be found in Material and Methods.

### *Closeness centrality* supports findings by expert curators

To confirm the relevance of closeness centrality, we applied this criterion to determine the optimal cutoffs in published SSN analysis by several independent research groups, based on real protein family datasets, to compare with the cutoff determined by expert curators. We created the SSNs of one Interpro family, three SFLD families, and two CAZy glycoside hydrolase families.

We first compared the closeness centrality performance to an analysis that used the popular SSN tool EFI-EST [5]. We selected one among its most recent citations in PubMed (September 2020), and with easily reproducible methods [10]. In this publication about the domain IPR037434, Travis *et al.* identified three relevant cutoffs including $10^{-55}$ corresponding to 5 subfamilies with more than 3 members. We generated SSNs from $10^{-5}$ to $10^{-100}$ (steps of $10^{-5}$), on which *closeness centrality* computations show a local optimum for an e-value of $10^{-60}$ (difference possibly influenced by the increase in the database size; 228 sequences in Travis et al., 376 in this investigation), which corresponds to 10 families with more than 3 members (51 subfamilies total). A consensus probably resides between those two.

For SFLD family 159 of tautomerases [11], the authors concluded first on an e-value of $10^{-11}$ for subfamily assignment, but due to the great diversity in their sequences also further investigated an e-value of $10^{-18}$ to define 18 subfamilies. The closeness centrality computations show local optima at e-values of $10^{-9}$ and $10^{-18}$, the latter displaying 15 subfamilies with >100 members. However, an absolute maximum was obtained at $10^{-34}$, although this would result in 12 more subfamilies (with >100 members), and more than 2000 smaller networks. For SFLD subfamily 19 of glutathione transferases [12], two cutoffs were identified by the authors, $10^{-13}$ and $10^{-25}$, the second level allowing more detailed inspections. The closeness centrality computations for this family peak at $10^{-14}$ and $10^{-29}$, with an absolute maximum at $10^{-44}$. For SFLD family 122 of nitroreductases, the published cutoff of $10^{-18}$ [13] also corroborates with the first local optimum of closeness centrality computations at $10^{-19}$. The global optimum was still not reached at $10^{-69}$, where we stopped our computations.

For our own recent SSN analysis of family GH16 [8], an e-value of $10^{-55}$ was published (SSNs generated in increments of $10^{-5}$), while closeness centrality revealed a peak at $10^{-50}$. Similarly, for GH130 [14], the authors determined the separation at $10^{-70}$, corresponding to 15 relevant subfamilies (>20 members). Closeness centrality reaches a peak at $10^{-75}$, resulting in 16 such relevant subfamilies. The subfamilies in CAZy are created according to the numbering of [14], with the omission of families purely derived from metagenomic data (UC4 and UC6) replaced by additional subfamilies we discovered (GH130_10 and GH130_12).

### *Closeness centrality* accurately points to suitable novel subfamily schemes

Closeness centrality was therefore applied to analyze SSNs generated for several CAZy families for which no subfamily scheme was designed yet. The family selection was based on potential interest for on-going projects, as well as on the diversity and total number of functional annotations available. Functional annotations are essential indicators to decide which scheme will be the most relevant and durable, among the several optimal options suggested by closeness centrality.

We first investigated the small GH55 family (1052 sequences after fragment filtering). This family also display low functional diversity as all characterized members breakdown b-

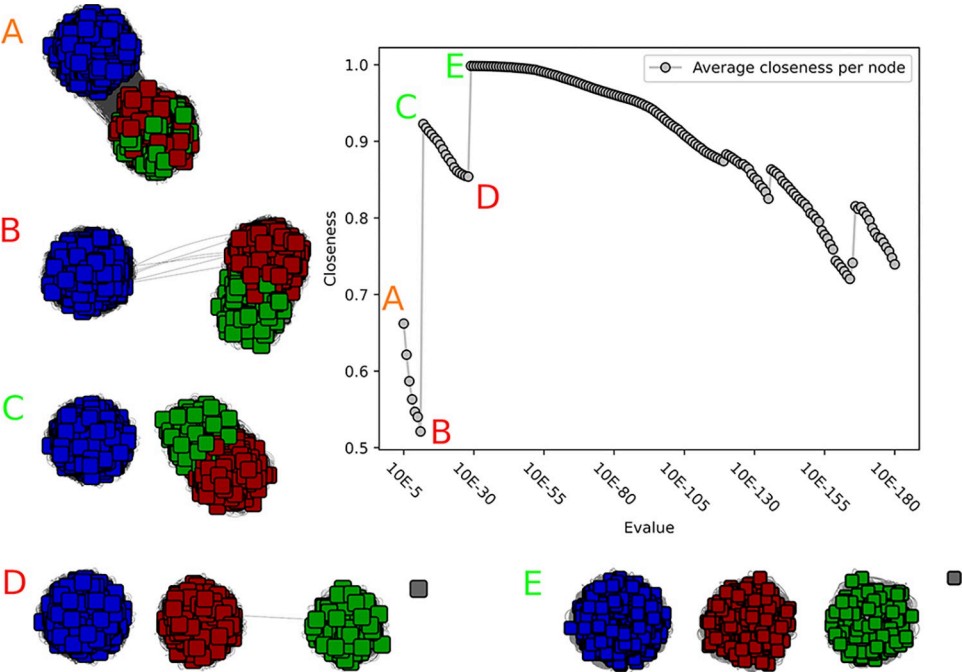

**Fig 2. Closeness value and corresponding networks of the GH55 SSN analysis.** A) At $10^{-5}$ the network is still a single network, and subnetworks become visible. The closeness value is in a medium range. B) The closeness value reaches its lowest point at $10^{-11}$. The two subnetworks within GH55 are only sparsely connected. C) At $10^{-12}$ the two subnetworks are disconnected, and the closeness value increases. D) The closeness value reaches another low at $10^{-28}$, and two other subnetworks are clearly visible, only sparsely connected with a single edge. A single disconnected node has emerged. E) At $10^{-29}$ three nearly perfect subnetworks are visible, and the closeness value is nearly perfect with a value of 0.99. An overview over the distribution of experimentally validated enzymes can be seen in S2 Fig.

1,3-glucans (mainly tested on laminarin), with the activity only differing by the mode of action: exo (EC 3.2.1.58), endo (EC 3.2.1.39), or undetermined (appearing in CAZy as laminarin-degrading enzymes, EC 3.2.1.-). Many characterized enzymes were described by Bianchetti and collaborators [15] who hypothesized that most members should be exo-acting. In their analysis of 177 GH55 proteins, the reconstructed phylogeny suggests three clades: a fungal, a bacterial and a divergent (bacterial) clade. The closeness centrality values for the generated SSNs indicated a local optimum at $10^{-12}$ and the global optimum, close to maximum, at $10^{-29}$ (Fig 2). At $10^{-12}$, two subfamilies emerged, one corresponding to the bacterial clade, the second gathering the fungal and "divergent" clades previously described. At $10^{-29}$, the fungal and divergent subfamilies split, resulting in a nearly perfect closeness centrality value. The two distinct modes of action did not segregate among the three subfamilies, given both are retrieved in the bacterial and in the fungal subfamilies, while the divergent subfamily does not include any characterized representative. As a consequence, GH55 will be divided into subfamilies 1 to 3, by Descending size order, which correspond respectively to the bacterial, fungal and divergent subfamilies.

## Multiple levels of subfamilies are possible

We investigated another small CAZy family, GH68 (1510 sequences after filtering). Three enzymatic activities were reported in this transglycosidase family (i.e. able to not only breakdown but also assemble glycans depending on the concentration of the substrate and product of the reaction): invertase (b-fructofuranosidase; EC 3.2.1.26), inulosucrase (sucrose:fructan

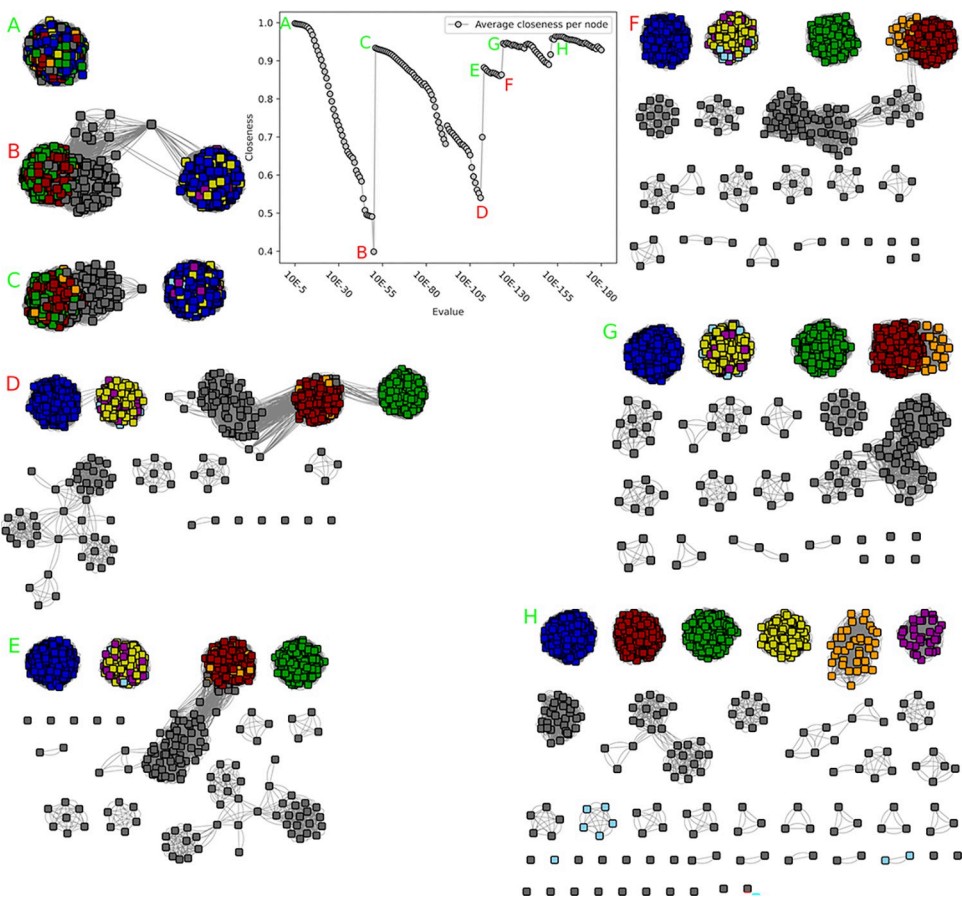

**Fig 3. Closeness value and corresponding networks of the GH68 SSN analysis.** Unlike in the GH55 analysis, the best network is less well determinable. A) At $10^{-5}$ the network is one solid component, and no subnetworks are visible. B) at $10^{-50}$ the closeness value is lowest, and two sparsely connected networks are visible. C) At $10^{-51}$ these two networks are disconnected, and the closeness value increases. D) to H) show the closeness values and networks at $10^{-111}$, $10^{-113}$, $10^{-123}$, $10^{-124}$ and $10^{-154}$ respectively. D and F show undesirable connections with sparely connected subnetworks, whereas E, G and H are alternative desirable network configurations which could be used to determine subfamilies. An overview over the distribution of experimentally validated enzymes can be seen in S3 Fig.

1-fructosyltransferase; EC 2.4.1.9) and levansucrase (sucrose:fructan 6-fructosyltransferase; EC 2.4.1.10). In this family, closeness computations never reached the initial maximum of the fully-connected network ($10^{-5}$). However, several local optima could be considered notably at $10^{-51}$, $10^{-124}$ and $10^{-154}$ corresponding to closeness centrality values of 0.93, 0.95 and 0.96 (Fig 3C, 3G and 3H). At $10^{-51}$, the family would be divided into two subfamilies of similar size. At $10^{-124}$, both former subfamilies get split into two (all having more than 150 members) while only one subfamily with >20 members emerges (plus 17 tiny groups including 5 single-tons). At $10^{-154}$, two more large groups emerge from the former four families. At the functional level, at $10^{-51}$, the first subfamily gathers EC activities 2.4.1.9 and 2.4.1.10, whereas the second displays 2.4.1.9, 2.4.1.10 and 3.2.1.26 (including one protein with dual activities 2.4.1.9 and 2.4.1.10, as well as one with dual 2.4.1.10 and 3.2.1.26). At $10^{-124}$, each of the five main subfamilies include a characterized member, but most activities are shared between these subfamilies, for example with 2.4.1.10 found in subfamilies 1 to 4; 2.4.1.9 in subfamilies 2, 4 and 5; and 3.2.1.26 in subfamilies 2, 3 and 5. At $10^{-154}$, 9 of the 19 subfamilies include a characterized member, five of which display a single activity. However, this activity is not unique but shared

(see S1 Table), given the split is mostly driven by the taxonomy, as at $10^{-124}$. Therefore, GH68 will be divided into two subfamilies following $10^{-51}$ threshold, based on the absence of functional information gain, and the increasing number of unclassified sequences at lower optimal thresholds (none at $10^{-51}$ versus 74 and 96 at $10^{-124}$ and $10^{-154}$ resp., that is >6%).

We then analyzed the larger (>10,000 sequences after filtering) GH51 family. This family displays so far four different EC numbers on a total of 83 characterized enzymes, 76 sharing the same a-L-arabinofuranosidase activity (EC 3.2.1.55), five with an endo-b-1,4-glucanase activity (EC 3.2.1.4) and two bifunctional enzymes: an a-L-arabinofuranosidase/xylan 1,4-b-xylosidase (EC 3.2.1.55 and EC 3.2.1.37) and an endo-b1,4-glucanase/endo-b-1,4-xylanase (EC 3.2.1.4 and EC 3.2.1.8). The *closeness centrality* criterion showed again multiple possible optimal SSNs, at $10^{-58}$, $10^{-91}$ and $10^{-147}$, corresponding to closeness values of 0.71, 0.67 and 0.79 that are lower than previously analyzed families (Fig 4A, 4C and 4D). At $10^{-58}$, the network divides into two large subnetworks (5712 and 3960 sequences) and one small network (114 sequences), hereafter referred as to subfamilies 1 to 3, and 55 smaller subnetworks (<100 sequences). Both subfamilies 1 and 2 contain enzymes with EC 3.2.1.55 activity, and subfamily 2 also contained the bifunctional enzyme EC 3.2.1.55 and 3.2.1.37. Proteins in subfamily 2 often display a small conserved domain of unknown function (annotated in the Gene3D superfamily 2.60.120.260) in the N-terminal region, unlike the enzymes in subfamily 1, further supporting the split between these groups. Subfamily 3 contains all five characterized enzymes with the EC number 3.2.1.4, including the bifunctional enzyme with EC 3.2.1.4/3.2.1.8. Most enzymes in subfamily 3 also contain a CBM domain from families known for binding cellulose (CBM2, CBM11, CBM30), which are not present in subfamilies 1 and 2. A small subfamily (55 sequences) also showed a different domain architecture as most members contain a CBM66 module, and some are fused to a GH127 enzyme. These features are not seen in any of the other subfamilies.

At $10^{-91}$, subfamily 1 and 3 split into two subfamilies, both containing characterized enzymes with similar activity/EC numbers. The split of subfamily 3 seems driven by the phylogeny, and resulted in 139 unclassified sequences as result of the split of subfamily 1. As a consequence, based on e-value range and functional specialization, 12 subfamilies of GH51 were build according to the $10^{-58}$ cutoff discussed above, including the three main subfamilies and nine small subfamilies among the 55 (having more than ten members; total 216 sequences) without functional evidence so far.

## Iterative SSNs might be necessary

The investigation of family GH45 initially resulted in 2467 modules sequences after fragment filtering. This only corresponded to a reduced fraction (77%) of the 3208 proteins annotated as GH45 and in-depth analysis revealed that the GH45 HMM did not allow to extract complete modules in many cases. Such a situation might result from extreme sequence diversity within a family, for which a single model will necessarily be highly degenerated on the most variable regions. An initial SSN was thus built on the 2467 proteins that passed the filter, and the closeness criterion used to determine a first optimal split at $10^{-7}$ allowing the generation of two subgroups. Two HMMs were built (one for each subgroup) and used to successfully extract 3200 more complete modules (passing the filter) among the 3208 GH45 sequences. Therefore, after one iteration, a second SSN was built, and analyzed with the closeness criterion.

There are three known activities in family GH45, namely endoglucanase (EC 3.2.1.4), xyloglucan-specific endo-β-1,4-glucanase (EC 3.2.1.151) and endo-β-1,4-mannanase (EC 3.2.1.78). Notably 13 enzymes display dual 3.2.1.4/3.2.178 activities, one enzyme dual 3.2.1.4/3.2.1.151, and another enzyme dual 3.2.1.78/3.2.1.151. A previous analysis identified 3 subgroups for this

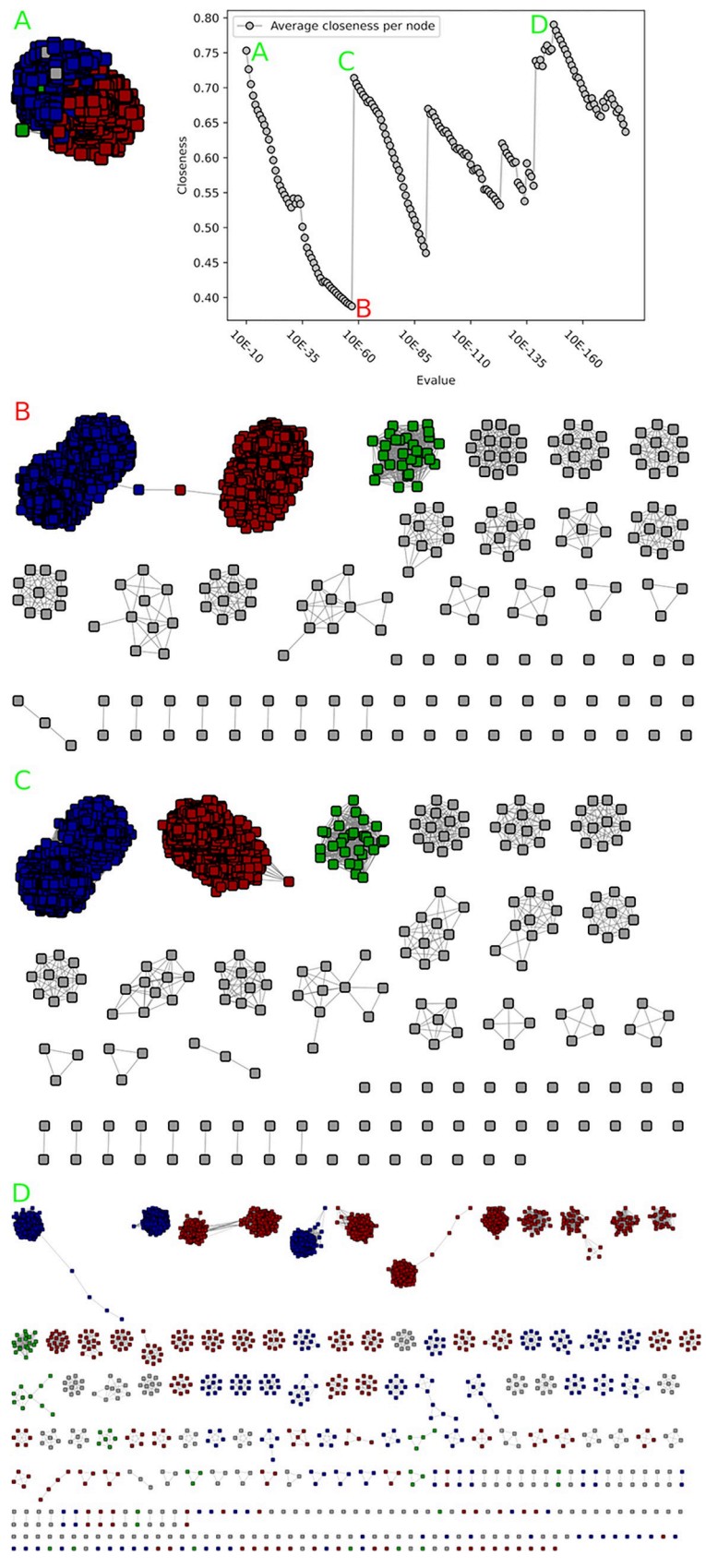

**Fig 4. Closeness value and corresponding networks of the GH51 SSN analysis.** As with GH68, multiple configurations could be of interest. A) The network is still only a single component. B) At $10^{-57}$, two main groups are sparsely connected, and the closeness value reaches its lowest point. C) At $10^{-58}$ these networks are disconnected, and the closeness value increases. D) $10^{-147}$ represents the highest peak after $10^{-58}$, and many more possible subfamilies are visible. All represented networks have been scaled down to 25% of their nodes, to make visualization possible. The network of A) was scaled down to 10%.

family [16]. Closeness centrality identified, after the initial $10^{-6}$ cutoff, another single additional optimum at $10^{-26}$ (Fig 5). At this cutoff, four major groups (>100 sequences; one with approx. 2000, two of ~440–510 and one of ~240) were identified, corresponding to the previously identified three groups [16] and an additional group. This new group (~440 members) consists nearly exclusively of Basidiomycetes sequences, which are globally more distant from other GH45 sequences, and has no characterized member yet. In the previous publication by [16] this group had not been seen, due to the small number of available Basidiomycetes sequences (only one sequence present in the previous analysis; XP_761686 in group C). This group did not appear in the first SSN, since the initial HMM did not detect a domain in all of its sequences. The sequences caught in the second SSN round were not limited to this group but also distributed over other small groups, therefore showing that the iterative SSN expanded multiple groups. No separation of EC numbers could be seen, with the largest group containing all three EC numbers, and the two smaller groups containing only the EC 3.2.1.4. Even at low e-values, e.g. $10^{-111}$, no full separation of the three activities could be obtained.

## Functional versus structural separation into subfamilies

Family GH19 was investigated, following a recent publication about its putative subfamilies [17]. We obtained 11,290 sequences after filtering, with in total two different activities, chitinase (EC 3.2.1.14) and lysozyme (EC 3.2.1.17) across 97 characterized enzymes. Closeness centrality identified multiple possible optima for subfamily separation (Fig 6). The first optimum

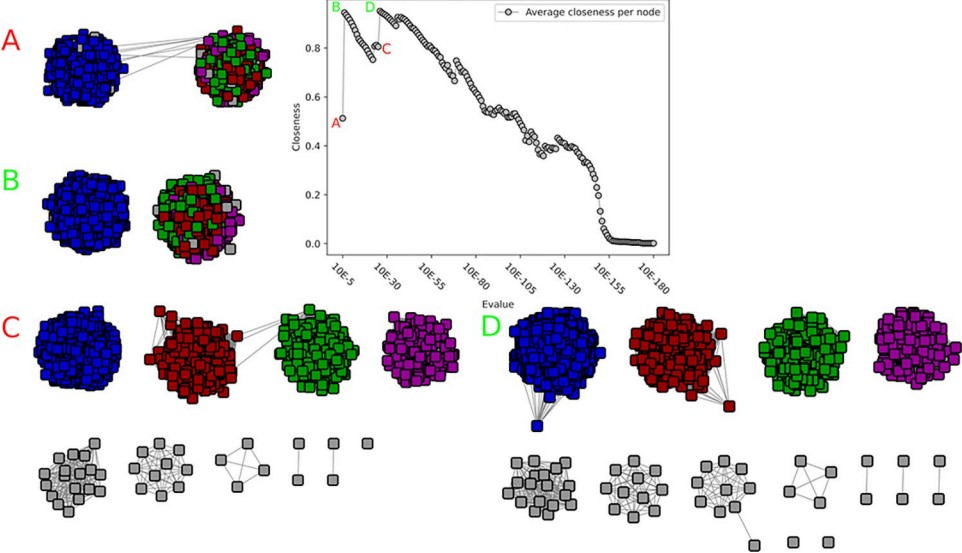

**Fig 5. Closeness value and corresponding networks of the second GH45 SSN analysis.** A) At $10^{-5}$ the network is still one component, although sparsely connected. B) At $10^{-6}$ the two subnetworks are disconnected and the closeness value increases. C) Another subnetwork has emerged, and two more are visible. D) At $10^{-26}$ the closeness value reaches its absolute maximum at 0.95 and four major subnetworks are visible. An overview over the distribution of experimentally validated enzymes can be seen in S4 Fig.

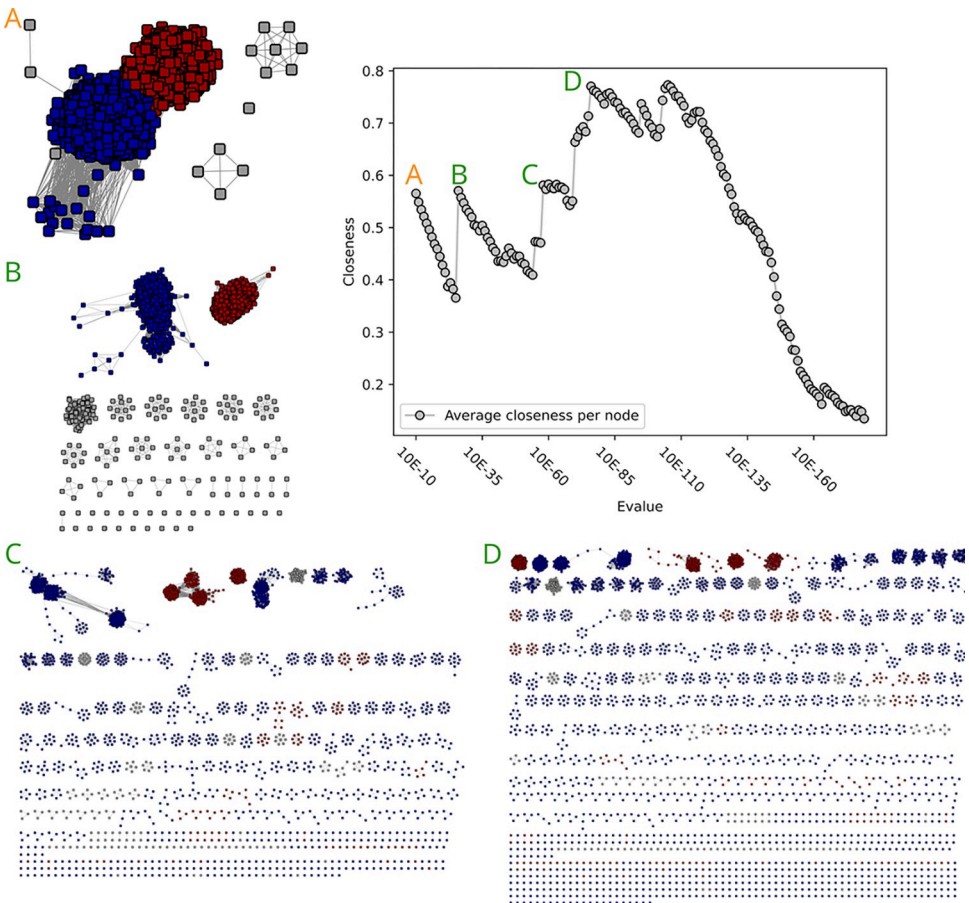

**Fig 6. Closeness value and corresponding networks of the GH19 SSN analysis.** As with GH68 And GH51, multiple configurations could be of interest. A) The network is still mostly a single component. B) At $10^{-26}$ two main networks are visible. C+D) $10^{-58}$ and $10^{-76}$ are other optima, showing a vast amount of subnetworks. All networks have been scaled down to 10% for visualization purposes.

at $10^{-26}$ (closeness of 0.57) splits the family into two main subgroups which display a single and specific activity, as well as 5 smaller subfamilies ($> = 10$ sequences) and 0.9% unclassified sequences. Some lower e-value cutoffs lead to higher values of closeness centrality ($10^{-58}$ reaches 0.58; $10^{-76}$ reaches the global optimum with 0.77) and would theoretically result in a better subfamily separation. It however leads to much more subfamilies (62 and 88 total with more than 10 members resp.), much more unclassified sequences (7% and 13% resp.), while no further separation of functions was observed, since known functions are already optimally split at $10^{-26}$. As discussed in [17], deeper subfamily splits result in a separation of structural features, and not of functions. We therefore decided to split GH19 into only two subfamilies.

## Not all protein families have subfamilies

While we were able to detect subfamilies for several families, some did not seem to display subfamilies based on the available data, such as families PL26 and GH54. For PL26 we obtained 1657 sequences after filtering, with only one characterized enzyme (EC 4.2.2.24). The closeness centrality value started perfectly at 1 for an e-value threshold of $10^{-5}$ and decrease very slowly up to 0.95 before a local optimum (closeness back to 1) at $10^{-46}$, where one small group of 21 sequences emerged (Fig 7). This group did not contain any known activities, and consisted out

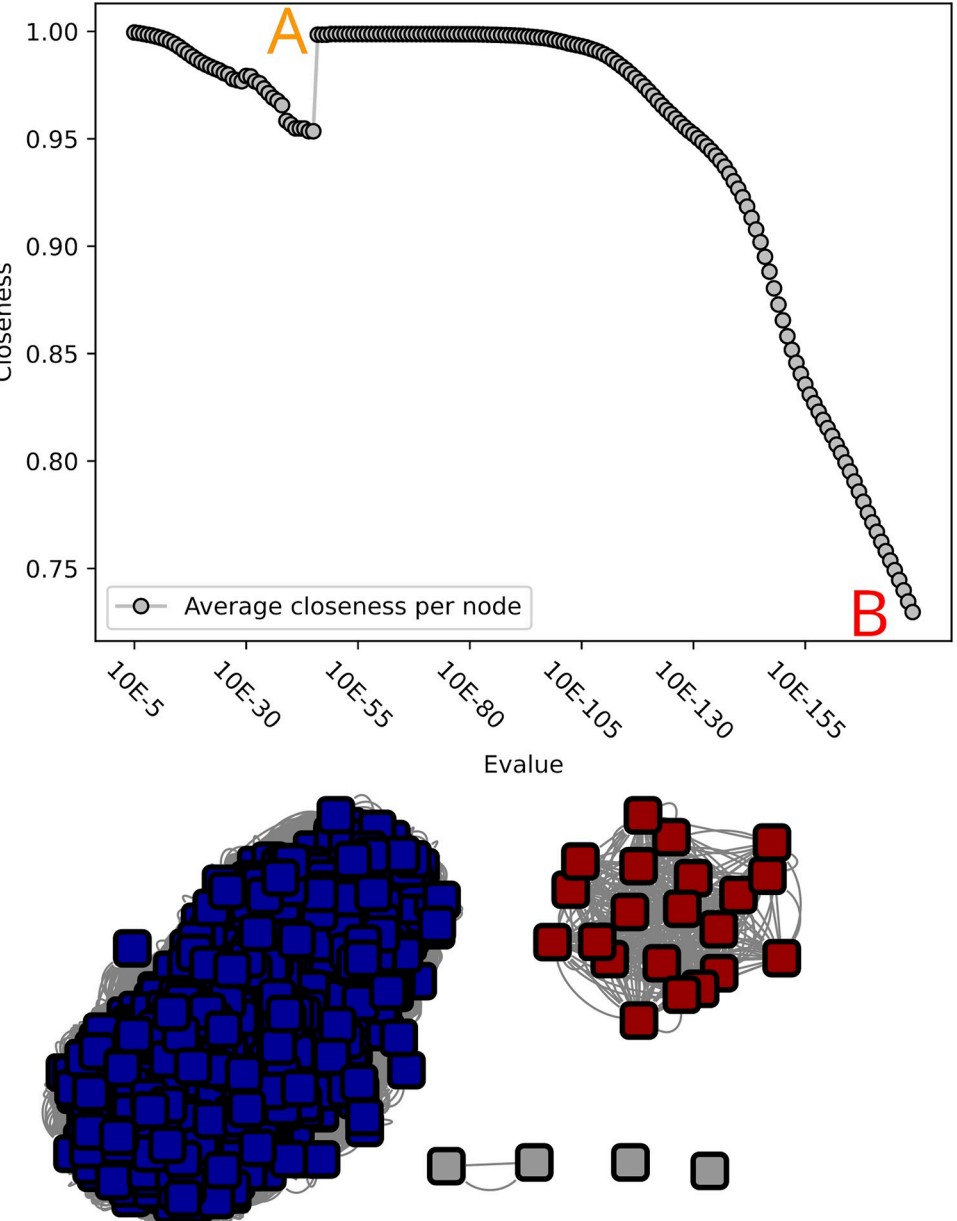

**Fig 7. Closeness value and corresponding networks of the PL26 SSN analysis.** A) The small red network is split off the main blue network B) Final network as depicted below the graph, at an evalue of $10^{-179}$.

of 20 sequences from the phylum Bacteroidetes and one of the phylum Balneolaetoa. At the highest e-value, $10^{-179}$, the PL26 network would still consist out of one main subnetwork of 1632 nodes, the small subnetwork of 21 nodes that appeared at $10^{-46}$, two linked sequences and two singles.

For GH54 we obtained 1594 sequences after filtering, including 20 sequences with known activities (19 with α-L-arabinofuranosidase, EC 3.2.1.55; one with dual α-L-arabinofuranosidase and β-xylosidase, EC 3.2.1.37 activities). In this case, closeness centrality also starts at 1 for the highest e-value cutoffs, stays stable until $10^{-67}$ (closeness centrality of 0.99) and slowly decreases until $10^{-131}$ (closeness of 0.70) before raising back to a local optimum at $10^{-139}$

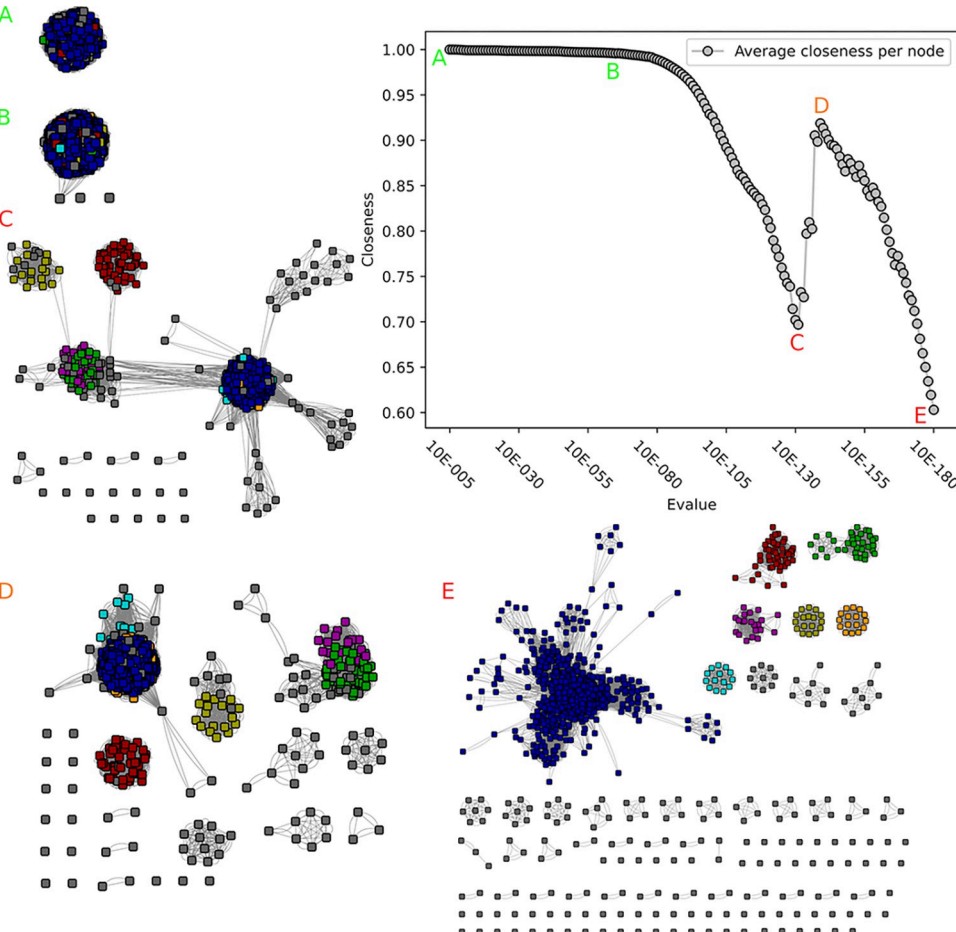

**Fig 8. Closeness value and corresponding networks of the GH54 SSN analysis.** A) The network is a single component, no subnetworks are visible. B) The closeness value changes slightly, and two disconnected nodes emerge. C) The closeness value reaches its first local minimum. While subgroups are visible, they are all still connected. D) The closeness value reaches its only peak. The main network still contains 86% of all sequences, and the smaller networks all follow phylogeny. E) At the maximum evalue of $10^{-179}$ various smaller networks emerge, yet no sensible bigger subnetworks are visible.

(closeness of 0.92; Fig 8). At this lowest cutoff, the largest subnetwork contains 1376 out of 1594 sequences, while the 216 remaining sequences spread over 25 smaller subnetworks (maximum size 86; only four with more than ten sequences). These subgroups follow taxonomic splits, and no functional division could be seen, since all characterized enzymes were contained in the main network.

We therefore concluded that based on the available data, and given the absence of characterized members among emerging small groups, no subfamily should be created for both PL26 and GH54.

## Discussion

The goal of this work was to identify an objective and performant criterion to facilitate the exploration of SSNs by expert curators during an analysis of subfamily delineation. We demonstrated on artificial data that the *closeness* centrality criterion displays encouraging behavior, given that it favors perfect subnetworks, and penalizes sparsely connected networks, which is

exactly what is desired for protein subfamilies translated into a SSN. We showed that the suggested optimal values are extremely close to those manually chosen by expert curators in previously published analyses, who sometimes identified several possible thresholds/subdivisions based on phylogenetic levels. In most cases, closeness centrality also identified several optima.

In the analyzed families, optima of lowest e-values reflected the taxonomic separation of the sequences and likely strict orthologous groups, while more functionally-relevant splits were obtained at higher thresholds. This likely reflect the evolution of proteins with ancient duplication leading to neofunctionalization bringing fitness to various surviving species, while taxonomic groups are accentuated by species survival/extinction, and sometimes by a bias in genome sequencing. Highest optima could correspond to more important structural modifications across family members, which might, or not, be associated to functional shifts. For example GH19 display similar activities despite structural variations, while a publication on GH128 suggests structural changes accompany substrate adaptation [17]. The existence of several optima still raises the question of the limitation of defining a single level of "subfamily". A more general framework with flexible number of sub-, subsub-, or more levels of classification might be relevant to have. This would allow (i) a better coverage of the sequence space, as many sequences cannot be assigned to a subfamily at lower thresholds; and (ii) finer functional prediction, associated to the diversity of the functions in the subdivision, hence broader at higher thresholds. In some cases, this might also depend on the scope of the database. For the specialized GH19 database [18], the authors chose to annotate multiple levels of subfamily, corresponding to first functional and later structural differences. The CAZy database is primarily interested in the functional differences, and we therefore decided that one subfamily-level annotation is sufficient, although this might depend on the families or enzyme classes. Therefore, we only implement one subfamily hierarchy level, as do many other specialized databases like MEROPS [19], ESTHER [20], or SulfAtlas [21]. The subfamily division is however not applicable to all families in a systematic/automatic manner, as it requires in-depth analysis to ensure long-term stability of the proposed classification, the availability of enough characterized members and human decision when several optima are found. This highlights the need to support such expert databases, which are dedicated to mine the functional annotation of (meta)genomes in the literature, as such effort can no longer be supported by universal databases.

For GH55, we found an agreement to prior analysis [15]. Bianchetti et al. identified, in a large-scale screening of GH55, three major subfamilies. Two of these families contained functional enzymes, and were divided into a fungal and a bacterial family, with the third family containing non-functional bacterial proteins. These three families correspond to the three groups identified by our SSN investigation, although we did not further subdivide based on phylogeny, as Bianchetti et al. did [15]. In this case we cannot only show that SSNs are able to delineate functionally different enzymes, in some cases they also might be able to delineate non-functional enzymes, as seen by the identification of a subfamily where no functions could be found even after testing. This potentially allows to save time in future endeavors by ruling out known non-functional proteins.

Our analyses also globally agreed with prior investigations of families GH68 [22], GH51 [23] and GH45 [16]. In case of GH68, this family would separate into two subfamilies and does not result into any unclassified sequences. A possible further in-depth classification, as also suggested by Velazquez-Hernandez, could be considered in the future, in case new functions are discovered in this more distant and small groups.

For GH51, three main subfamilies, corresponding to the three main clades identified in [23], will be created, as well as nine smaller ones. The two biggest subfamilies, representing 96% of the data, are distinguished by a structural feature in the sequences. One protein in one

of the subfamilies contains a minor side function. Further biochemical characterization is necessary to see if this is a property of the whole subfamily and if it might be a distinguishing factor between the two major subfamilies. The third major subfamily represents a distinct function, whereas the fourth is of unknown function, but is characterized by the presence of a carbohydrate binding domain, indicating considerable differences within this family.

The GH45 case differs from the previous cases as none of the subfamilies demonstrated separate biochemical functions. Yet four subfamilies will be implemented, since this family seems to be very diverse, and the initial HMM for the detection of GH45 did not allow a full detection of module sequences. The subfamily HMMs yield an improved performance and better detection, making their implementation therefore necessary.

For some families, we were not able to identify any sensible subgrouping. We hypothesized that two or more groups could exist in GH54, because most enzymes in GH54 contained only one function, but one enzyme from *Trichoderma koningii* has been deemed to have a dual activity (additional β-xylosidase activity) [24]. The SSN analysis indicated that no such major subgroups exist though. While reviewing the literature, it appears that this β-xylosidase activity is only a minor side-activity, which also was reported for other enzymes in this group before [25,26]. We therefore conclude that no subgroups exist in GH54, and that potentially all members might have a dual activity.

Family PL26 was included in our analysis, since it is the largest PL family, for which no subfamilies were previously created [27]. After the creation of the SSN, we were unable to detect any major subfamilies. Given that only one protein has been characterized so far, it might be that this family truly does not contain any more functions or subfamilies.

A limitation of the *closeness centrality* is that it is independent of the amount of subnetworks and of their size. Two perfect subnetworks of the size 50 will give the same outcome as 50 perfect subnetworks of size 2. An option would be to penalize subnetworks below a certain size, typically below the required family size, by downgrading their values e.g. assigning a value of 0 as for singlets. While this might be necessary in case many different networks are built at the same time, in most circumstances a manual evaluation by the curator with the amount of networks in mind should suffice.

An unexpected limitation which we encountered is the usage of the e-value for the generation of SSNs. Most researchers decrement the e-value to generate their SSNs [4,5]. The e-value is a floating-point value, where reaching 0.0 is considered to be the most significant value, indicating a very good match, most often resulting in perfect matches. The e-value is dependent on database size and query length though. In case of too large families, or very long protein domains (e.g. PL26s average is 867 AA), the e-value will often result in 0 (see S5 Fig, with an overview of e-values per percentage identity), due to the limit of $10^{-308}$ for a 64-bit floating point value with double precision (IEEE 754 standard for floating point arithmetic, [28]; the equivalent limit for 32-bit is $10^{-38}$). Distribution of e-values can often be seen with a maximum of $10^{-160}$ or $10^{-180}$, with more significant values being 0 for long domains. For these long domains this can lead to many hits, which identity percent can reach fairly low/limit values between 20–30%, and yet lead to an e-value of 0. In case of PL26, e-values of 0 were obtained at worst with a sequence identity of 32%. Bitscores (or a combination of identity percent and coverage control), should rather be used instead of e-values in case of unusually long sequences.

We also tried to evaluate a proposed mechanism for network filtering from the field of neuroscience [29], but this proved to be computationally infeasible, with runtimes of more than 3 days for comparatively small networks from our collection. We furthermore hoped to aid our annotation efforts with well-known algorithms for grouping data points, like clustering or PCA, despite them not being favored in earlier research [30]. It turned out that even for rather simple group combinations with rather little sequences, like two or four groups in GH68, these

algorithms are already severely challenged (see S1 Text). At this point it needs to be evaluated if the results from the other techniques like PCA are correct, and the results based on *closeness centrality* might be incorrect. A manual inspection with Cytoscape [31] proofed to be efficient and simple for the small SSNs, and confirmed that the results obtained by the evaluation of *closeness centrality* are indeed correct. Overall, these results from the other techniques make our exploration of other criteria even more worthwhile. We hope that the *closeness centrality* metric will be used during other research projects in this field, to combine expert knowledge with reproducible technical measures.

## Materials and methods

### Generation of artificial data

Artificial data was generated with the Python networkX 2.4/5 library [32].

We first used the network generators to create different *subnetworks* (to be connected during the second step): (i) two perfect networks (all possible edges between all nodes exist) of size 50 and 200, using the *nx.complete_graph* function; six random networks of size 200 with 300, 500, 1000, 2000, 3000 and 4000 edges (representing from 1.5% to 20.1% of a complete graph) using the *nx.gnm_random_graph* function; and two networks consisting out of 50 or 200 single nodes, or singletons.

Secondly, we generated network *collections*. A *collection* consists in multiple *networks* build upon the same initial pair of *subnetworks* (described above; see S2 Table). A *collection* starts with a first *network* simply consisting in the two disconnected *subnetworks*. The collection is completed by additional *networks* obtained by randomly connecting the nodes of one *subnetwork* to the other with an increasing number of edges, from one to the maximum possible (see S6A Fig). This means that a combination between a *subnetwork* of size 50 and another of size 200 would result in a *collection* that contains 10,001 *networks*, one without connection and then from a single connection between a random node of each *subnetwork*, up to the connection all 50 nodes of the first *subnetwork* to the 200 nodes of the second *subnetwork*. In case of two perfect *subnetworks*, the lastly created *network* in the *collection* would also be a perfect network.

In the specific case of a single-node *subnetwork* combined to a random or perfect *subnetwork*, two additional strategies of connections were performed to better explore the space of topologies, while ending in the same number of *networks* in the *collection*: 1) all single nodes (initially one and lately all) were first randomly connected to nodes of the other *subnetwork*. Once all connected, a second edge was added to each of these nodes, and this was repeated until all connections were made (see S6B Fig); 2) a random node of the single-node *subnetwork* was first connected to all nodes the other *subnetwork* (initially to one and lately to all), and only afterwards a second single node was connected in the same way, until all connections were made (see S6C Fig).

Not all possible *subnetwork* pairs were used to generate *networks/collections*, as only the following cases were considered: perfect network of size 50 randomly connected to all possible *subnetworks* (single node of size 1, 50 and 200; perfect subnetworks of size 3, 50 and 200, random networks of size 50 with 250 and 500 edges, and random networks of size 200 with 500 to 4000 edges) and single node networks (size 50 and 200) connected to random networks of size 200 with 500 to 4000 edges with the three connection strategies. An overview is given in S2 Table.

### Network properties

Before the execution of any computation, the networks were converted to an undirected graph. Afterwards, various network properties were calculated, using networkX, and

investigated for local and global minima and maxima. These 19 properties include the number of components, edges, s-metric, average degree centrality, average closeness centrality (with the parameter *wf_improved*—allowing to cope with disconnected components—either True and False), average betweenness centrality, average degree, average clustering, eigenvector centrality, betweenness centrality source, edge betweenness centrality, load centrality, edge load centrality, subgraph centrality, harmonic centrality, global reaching centrality, the amount of local bridges, the shortest path and asyn_lpa_communities [33,34]. For the *artificial* data, all these properties were calculated, while CAZy families GH16 [8], GH19 [18], GH45, GH51, GH54, GH55, GH68, GH130 [14] and PL26, as well as different families of the Structure Function Linkage Database (SFLD) [3], SFLD families 19, 122 and 159 and Interpro IPR037434, were investigated with the closeness criterion only (see results). An overview can be seen in S3 Table.

## Closeness details

After evaluating the different parameters (see results), closeness centrality was chosen as the indicator for subfamily network optima. Closeness centrality [33] of a network corresponds to the averaged closeness centrality *C(u)* over all nodes *u* connected to at least another node *v*. The closeness centrality of a node is the reciprocal of the average shortest-path distance to all (other) reachable nodes. It is defined by the formula: $C(u) = (n-1)/\sum_v sp(u,v)$ where *sp(u,v)* is the shortest-path distance between nodes *u* and *v*, that is the minimal number of edges to cross to reach *v* from *u*. A value of *C(u) = 1* indicates the existence of an edge bridging *u* to all other nodes in its connected component, while lower values will indicate the isolation of *u*: absence of direct access, with increasing distances, to many nodes. A closeness centrality of a network thus suggests that the defined subfamilies are well-packed, while lowering values reflects the drift of some (groups of) proteins, away from the rest of their relatives. The closeness centrality will raise up when these (groups of) proteins finally separate to create another connected component. An example is given in S7 Fig.

The parameter *wf_improved* in networkX was set to *False*, to prevent the Wasserman and Faust correction, which was designed to consider the network as a whole, and thus penalizing closeness centrality value if the graph is composed of several subcomponents, which is exactly the objective of our study.

A script allowing the computation of closeness centrality from a given network file has been made available on GitHub under https://github.com/bastian-wur/closeness_by_component.

## Generation of SSNs

Sequence similarities on SFLD and CAZy families were computed with Blastp v2.5 [35]. Standard parameters were used, besides the parameter *max_target_seqs*, which was scaled to the size of the family/target database. The Blastp pairwise hits, based on e-value cutoff, were transformed into a digraph with the python library networkX v2.4/2.5 [32], while sequences without any hit were added as single nodes. We generated SSNs for each e-value cutoff, from $10^{-5}$ up to $10^{-160}$, by step of $10^{-1}$. Exceptions to these start- and end-points were made when either the network did not form any more subnetworks at high cutoffs, or when the network disintegrated into too many subnetworks (typically >1000) at low cutoffs.

All networks were visualized with Cytoscape version 3.7.2 [31]. Graphs showing the evolution of the network properties were generated with Matplotlib [36–38].

GNU parallel version 20161222 has been used during this and other steps of this study [39].

## Runtime performances

Computational time of the closeness centrality is mainly dependent on the number of edges. For the small GH55 network (1052 nodes, at max 30,2970 edges) run times per network were below 4 minutes. The whole network collection (175 networks; $10^{-5}$ to $10^{-179}$) could therefore be processed in 12 hours on a single core on a desktop computer, without parallelization. In general, most network collections with 5,000 or less nodes can be processed with 5–6 parallel processes in less than 48h on a desktop computer. For the biggest network, GH51 (10,086 nodes, at max 33,530,398 edges), runtimes were between 5 days (biggest network at $10^{-10}$) down to 45 minutes (smallest network at $10^{-179}$). RAM consumption was more than 120 GB RAM for the biggest network (therefore requiring a server), but is again negligible for the smallest ones.

Several optimization strategies could be considered, e.g.by only collecting first the information about the number of components, and then processing only a network if the number of components change, or by reducing the number of investigations, for example from $10^{-30}$ to $10^{-120}$.

## EFI-EST

The EFI-EST website [40] was used to reproduce the published IPR037434 family analysis, similarly to [10]. An initial e-value cutoff of $10^{-5}$ was used, and subnetworks were produced based on this network by decreasing the e-value cutoff by steps of 10–5 until the maximum selectable e-value of $10^{-100}$. Steps of $10^{-1}$ were prohibitive in an online-tool without the possibility of automation, and edge values were not included into EFI-EST SSNs during our first tests.

For reading xgmml files, networkxgmml v0.1.6. was used (https://pypi.org/project/networkxgmml/).

## Extraction of domain sequences

SFLD sequence data for families 19 of glutathione transferase [12], family 159 of tautomerase [11], and family 122 of nitroreductase [13] were downloaded from http://sfld.rbvi.ucsf.edu/archive/django/superfamily/index.html [9]. For these families the full protein sequences were used.

Data for glycoside hydrolase families GH16, GH19, GH51, GH54, GH55, GH68, GH130 and polysaccharide lyase family PL26 were directly extracted from the CAZy database [41], and were further supplemented with data by collaboration partners. Domain sequences were extracted from the full proteins using in-house pipeline based on the results of *hmmscan* function in HMMer3.3 [42] as follows. HMM searches using the in-house CAZy HMM profiles were performed and only matches with an e-value <1–4 were further processed. The sequence and HMM coordinates were extracted in an attempt to reconstruct a domain sequence from fragments created by the local detection of HMM searches. Two consecutive fragments were assembled only in the given conditions: (i) sequence start coordinates must be separated by >20 amino-acids, as well as sequence end coordinates; (ii) sequence end of the first fragment should not be separated from the start of the second fragment by >200 amino acids; (iii) HMM coordinates must not overlap (end of the first vs start of the second) by more than 30 amino acids. Any domain sequence, initially selected or reassembled, shorter than half of the HMM length was discarded. A manual inspection was performed for sequences longer or shorter than the average sequence length +/- three standard deviations, to discard gene models issues. In all cases >95% of the original input sequences passed the screening, except for GH45 (see Results section). For the construction of the second SSN for GH45, domains were

extracted as described but using two GH45 HMMs. In case both HMMs identified a domain in the same protein, the longer domain was used.

## Minimized version for visualization

Since many of the generated networks were too large to be visualized in Cytoscape, and preliminary clustering seemed to warp network structure in some cases, networks for visualization were generated differently. A custom and modified version of uniform edge sampling with graph induction was implemented [43], to generate representative smaller graphs from the full graph.

To retain network structure, subgraphs were generated per connected component. Components smaller than a chosen node amount (in most cases 10) were retained as they were. Larger components were sampled down to a specified percentage (in most cases 25 or 50%; exact values specified per figure), with respect to the minimum amount of nodes of small components. The subsampled graphs, initially empty, were completed by random edges (Python *random.choice* function) and their nodes sequentially until the minimum amount of nodes was reached. At last, all edges connecting the sampled nodes in the initial component were added as well [43]. In case a component was split by the random subsampling, reconnection was realized as follows. Iterating over all split sub-components, the shortest path algorithm (as implemented in networkX) was used to determine the shortest path between these selected nodes based on the initial component, and all necessary edges were iteratively added. The computation was stopped when the component was again fully connected.

## Supporting information

**S1 Table. List of artificial networks used for the selection of a valid criterion.**
(XLS)

**S2 Table. List of ECs per GH68 subfamily, depending on evalue cutoff.**
(XLS)

**S3 Table. List of SSNs cutoffs computed in this and prior works.**
(XLS)

**S1 Text. Supplementary Methods and Results, additional methods and results not described in the main manuscript.**
(DOCX)

**S1 Fig. Behaviour of different metrics over two networks the size of 50.** A) Closeness and degree centrality both favour more edges over less, and both networks combined with zero connections score worse than a network with one connection. The adjustment to score these networks better has been performed in this work. B) The number of edges, the s-metric, the average degree, subgraph centrality and harmonic centrality show a monotonous behaviour. The more edges a component possesses, the better the score. This means that one perfect network scores worse than two perfect networks with one connection. C) Betweenness and similar metrics score in an inverted way to closeness in A). Based on this behaviour it cannot be decided which metric is more suitable. A decision had to be made based on the performance with single nodes, as discussed in the manuscript. D) The amount of bridges, components and communities does not show a quantitative enough behaviour. Many instances are scored with a value of one, making it not possible to discriminate between better or worse networks. E) and F) show the behaviour of the remaining four criteria. The eigenvector centrality had to be plotted separately, without the first three data points, due to differences in scale between these

data points. All four metrics show a behaviour which might favour less edges over more edges at some point in the data set, making their behaviour not desirable. Parameters may have been scaled for display purposes.
(PDF)

**S2 Fig. Distribution of experimentally validated enzymes in GH55, depending on evalue cutoff. At $10^{-5}$ only one network exists with all 50 experimentally validated enzymes, which then further splits up into two networks at $10^{-12}$ and at the end into three networks at $10^{-29}$.** At $10^{-29}$ the first network of 699 nodes contains 32 experimentally validated enzymes, the second network of 248 contains 18, and the last network does not contain any characterized enzymes. The bottom row summarizes the amount of protein sequences not in a subfamily at the respective evalue.
(PDF)

**S3 Fig. Distribution of experimentally validated enzymes in GH68, depending on evalue cutoff.**
(PDF)

**S4 Fig. Distribution of experimentally validated enzymes in GH45, depending on evalue cutoff.**
(PDF)

**S5 Fig. Violin plot of percentage identity values matched with corresponding evalue values for PL26.** With 32% identity between two sequences the first evalue of zero was obtained. With 64% identity between two sequences all evalues correspond to zero.
(PDF)

**S6 Fig.** A) One network set, consisting out of two networks, which are connected with increasing amounts of edges. B) Special case #1: one network is connected to a network of singles. Each single node is connected first by only one connection, and not random, preventing a second connection before all nodes are connected. C) Special case #2: one network is connected to a network of singles. Each single node is connected first to all nodes in the main network.
(PDF)

**S7 Fig. A visualization of the closeness calculation.** In A) a perfect network is displayed. From the example of the grey node, all distances to other nodes (red, cyan, beige) is one. B) an imperfect network is displayed. From the grey node three distances are one (red, green, beige), one distance is two (cyan), and one three (purple). The average of these paths is used as the closeness of the whole network.
(PDF)

**S8 Fig. F1 score of the HMM subfamily assignments for GH68. 2 new subfamily HMMs were built, and the best F1 score with least FP and FN was seen if a new sequence was assigned to a subfamily if the HMM reported an e-value between $10^{-5}$ and $10^{-35}$.** The distance to the next best HMM hit was also considered, but choosing no difference (first line in legend) or choosing a minimum of $10^{-5}$ e-value difference to the second best hit (second line in legend) did not make any difference in this case (graphs overlap).
(JPG)

**S9 Fig. F1 score of the HMM subfamily assignments for GH51. 12 new subfamily HMMs were built, and the best F1 score with least FP and FN was seen if a new sequence was assigned to a subfamily if the HMM reported an e-value at 10E-45.** In addition, the distance

to the next best HMM hit has to be considered, since requiring a minimum of 10E-10 between the best HMM hit and the second best HMM hit gave the best result.
(JPG)

**S10 Fig. Example of dimensionality reduction results. Colouring according to groups identified by the SSNs analysis.** For GH68, both groupings at 10E-51 and 10E-154 are displayed. For the SFLD families, pre-computed groups were used. While in the PCA plot for GH55 the three subfamilies are visible, this gets ambigious in the MDS plot or for GH68, and indistinguishable for the SFLD families.
(PDF)

# Acknowledgments

The authors would like to thank Vincent Lombard for technical support and for administration of the CAZy cluster. The authors would furthermore like to thank Bernard Henrissat for proofreading and providing useful comments.

# Author Contributions

**Conceptualization:** Bastian Volker Helmut Hornung.

**Data curation:** Bastian Volker Helmut Hornung, Nicolas Terrapon.

**Formal analysis:** Bastian Volker Helmut Hornung.

**Funding acquisition:** Nicolas Terrapon.

**Investigation:** Bastian Volker Helmut Hornung.

**Methodology:** Bastian Volker Helmut Hornung, Nicolas Terrapon.

**Project administration:** Nicolas Terrapon.

**Resources:** Nicolas Terrapon.

**Software:** Bastian Volker Helmut Hornung.

**Supervision:** Nicolas Terrapon.

**Validation:** Bastian Volker Helmut Hornung.

**Visualization:** Bastian Volker Helmut Hornung.

**Writing – original draft:** Bastian Volker Helmut Hornung, Nicolas Terrapon.

**Writing – review & editing:** Bastian Volker Helmut Hornung, Nicolas Terrapon.

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
