## [Decision Letter · Decision Letter 0]

7 Jul 2022

Dear Dr Hornung,

Thank you very much for submitting your manuscript "An objective criterion to evaluate sequence-similarity networks helps in dividing the protein family sequence space" for consideration at PLOS Computational Biology. As with all papers reviewed by the journal, your manuscript was reviewed by members of the editorial board and by several independent reviewers. The reviewers appreciated the attention to an important topic. Based on the reviews, we are likely to accept this manuscript for publication, providing that you modify the manuscript according to the review recommendations.

Sincerely,

Rachel Kolodny

Associate Editor

PLOS Computational Biology

William Noble

Deputy Editor

PLOS Computational Biology

[LINK]

Reviewer's Responses to Questions

**Comments to the Authors:**

Reviewer #1: This paper demonstrated an interesting and useful approach for the SSN analysis, i.e., what is an appropriate e-value threshold to delineate different subfamilies (or subgroups). Indeed, there is no systematic method to solve this problem and researchers use arbitrary and intuitive cutoffs to generate own SSNs in general. The approach that the authors described in this paper offers a nice solution for this problem.

The paper flows well and well written in general. I believe that this paper has a great value to publish in PLoS Computational Biology while I have some suggestions to improve the manuscript.

1. While the authors presented the approach to measure several parameters for the SSNs, there is almost no description in the method. The general methods have been previously developed and the authors simply cite those papers in Methods. As this paper is really centered for the utility of the methodology, I request to describe step-by-step protocol that other researcher can replicate and adopt to other system. If they use own script, the authors must upload it to github to share it to the community.

It is always difficult/impossible to know if clustered sequences represent isofunctional subfamily or just clustered sequences (more subgourp) with potentially multiple sub-subfamilies. It is important to map and highlight sequences with experimentally characterized in the SSNs. I wonder if the authors can highlight those sequences for each family they have characterized. They can isolate those experimentally validated sequences from Uniprot and map onto their SSNs. Mapping those information gives much confidence for the utility of the approach.

Reviewer #2: Uploaded as an attachment

**Have the authors made all data and (if applicable) computational code underlying the findings in their manuscript fully available?**

Reviewer #1: **No: **

Reviewer #2: Yes

PLOS authors have the option to publish the peer review history of their article (what does this mean?). If published, this will include your full peer review and any attached files.

Reviewer #1: No

Reviewer #2: No

Figure Files:

Data Requirements:

Reproducibility:

References:

---

## [Editor Report · Decision Letter 1]

18 Jan 2023

Dear Dr Hornung,

We are pleased to inform you that your manuscript 'An objective criterion to evaluate sequence-similarity networks helps in dividing the protein family sequence space' has been provisionally accepted for publication in PLOS Computational Biology.

Best regards,

Rachel Kolodny

Academic Editor

PLOS Computational Biology

William Noble

Section Editor

PLOS Computational Biology

---

## [Editor Report · Acceptance letter]

29 Jan 2023

PCOMPBIOL-D-22-00646R1 

An objective criterion to evaluate sequence-similarity networks helps in dividing the protein family sequence space

Dear Dr Hornung,

I am pleased to inform you that your manuscript has been formally accepted for publication in PLOS Computational Biology. Your manuscript is now with our production department and you will be notified of the publication date in due course.

With kind regards,

Zsofia Freund
